# Effects of a simulated marine heatwave on the structure and composition of Mediterranean plankton in a mesocosm study

Zoé Eglaine[1]*, Justine Courboulès[1]¤, Francesco Cipolletta[2,3], Cécile Roques[1], Tanguy Soulié[1], Diana Sarno[3], Behzad Mostajir[1], Francesca Vidussi[1]*

**1** MARBEC (Marine Biodiversity, Exploitation and Conservation), Univ Montpellier, CNRS, Ifremer, IRD, Montpellier, France, **2** Settore Biologia Levante, Agenzia Regionale per la Protezione dell'Ambiente Ligure, La Spezia, Italy, **3** Department of Research Infrastructures for Marine Biological Resources, Stazione Zoologica Anton Dohrn, Naples, Italy

¤ Current address: Norwegian University of science and Technology, NTNU, Department of Biology, Trondheim, Norway
* zoe.eglaine@cnrs.fr (ZE); francesca.vidussi@cnrs.fr (FV)

## Abstract

Coastal marine systems are particularly affected by marine heatwaves (MHW), which affect organisms, including plankton communities, that are essential for ecosystem function and productivity. To study their effects on plankton food web components, an *in situ* mesocosm experiment was conducted in Thau Lagoon (Mediterranean Sea, South France) from May to June 2019. The two conditions were applied in triplicate. A MHW of + 3 °C above the natural lagoon water temperature was applied to three mesocosms and maintained for the first 10 days of the experiment. Afterward heating was discontinued, and temperatures returned to ambient levels for the remaining 10 days. The other three mesocosms were maintained at natural water temperatures throughout the experiment. Phytoplankton responded positively to MHW, whereas protozooplankton and viruses exhibited significant negative responses. The decrease in protozooplankton, which predominantly preyed on phytoplankton, can be explained by the increase in metazooplankton observed under MHW. Increased predation by metazooplankton on protozooplankton reduced their grazing pressure on phytoplankton, allowing them to proliferate. Simultaneously, metazooplankton directly grazed on larger phytoplankton cells, thereby reinforcing the shift in community composition towards smaller species following the simulated MHW. These combined top-down effects led to pronounced changes in both the structure and size of phytoplankton communities under MHW condition, driven by trophic cascades within the planktonic food web. Plankton functional group stability metrics showed that smaller communities were more resistant to MHW than larger communities. Nanophytoplankton, autotrophic flagellates, heterotrophic dinoflagellates, and tintinnids exhibited minimal recovery, whereas other plankton communities displayed a pronounced capacity for

**Data availability statement:** All data files underlying this study are available from SEANOE datbase (DOI:10.17882/108308).

**Funding:** The present research was funded under the AQUACOSM project, which have received funding from the European Union's Horizon 2020 Research and Innovation Program (H2020/2017–2020) under grant agreement n731065.TS grant was also partly founded from AQUACOSM and FC benefitted from AQUACOSM transnational access founds. Part of the microscopy analyses were done with the support of LabEx CeMEB, an ANR "Investissements d'avenir" program (ANR-10-LABX-04-01). The funders had no role in study design, data collection and analysis, decision to publish, or preparation of the manuscript.

**Competing interests:** The authors have declared that no competing interests exist.

full or near-complete recovery following MHW. The contrasting resistance, resilience, and recovery of planktonic functional groups to MHW led to a restructuring of the planktonic food web and its function, with potential consequences for key ecological processes in the pelagic ecosystem.

## Introduction

Marine heatwaves (MHWs) refer to periods of persistent high temperatures that occur in seawater at specific locations [1,2]. They are characterized by unusually warm seawater that exceeds a defined threshold and are typically described in terms of their duration, intensity, and spatial extent [1,3,4]. Between 1925 and 2016, MHWs increased in both frequency and duration by 34% and 17%, respectively [5]. Predictive models further suggest that MHWs will intensify and occur more frequently by the end of the century [2,6]. Coastal marine environments are particularly affected by MHW, leading to the abrupt redistribution of marine species, alterations in biological processes, such as elevated basal metabolic rates that may cause energy demands to exceed the metabolic capacity of species [7,8], and even mass mortality events [4,9]. These events cause decreased species diversity, shifts in species abundance and composition, and structural and functional changes in the microbial plankton food web, and altered plankton dynamics [10–12].

Microbial plankton communities drive marine food web dynamics, and play a key role in primary production and biogeochemical cycles (oxygen, carbon, phosphorus, etc.) [13–16]. These communities are shaped by complex top-down (e.g., predation, parasitism, viral lysis) and bottom-up (e.g., nutrient- and light-driven competition or mutualism) interactions [14]. The high diversity of microorganisms in the plankton food web, including viruses, bacteria, pico-, nano-, and micro-phytoplankton, and protozooplankton, contributes to this complexity. Protozooplankton, such as heterotrophic flagellates and ciliates, actively graze on bacteria and small phytoplankton, transferring energy to higher trophic levels, including metazooplankton [17,18]. Heterotrophic bacteria and phytoplankton play crucial roles in nutrient cycling in marine ecosystems [19]. Viruses also act as top-down regulators by lysing bacterial and phytoplankton cells, thereby contributing to nutrient recycling [19,20]. These dynamic interactions are essential for maintaining the ecosystem function and stability.

Temperature strongly influences the physiological and biological processes of marine species, affecting growth, reproduction, body size, feeding, and behavior [21–23]. These changes can have cascading effects across multiple organizational levels, from individuals to communities and ecosystems. MHW can affect community recovery by reducing either resistance (the ability to withstand change) or resilience (the ability to return to the original state) [24]. Coastal ecosystems, including lagoons, are more exposed to stressors than open marine ecosystems because of their shallow depths and limited water volumes. These characteristics make them particularly sensitive to atmospheric temperature fluctuations that can drive significant daily thermal variability and potentially promote faster and more flexible adaptive responses

to external pressures [25]. Moreover, the resilience of lagoon communities, supported by their heterogeneity and self-regulatory mechanisms, may surpass that of open marine ecosystems [25,26]. Understanding how MHW influences these processes and interactions is key to anticipating changes in microbial community composition and functions within marine food webs and ecosystems.

A widespread trend in response to warming is the increased dominance of small phytoplankton, such as pico- (0.2–2 µm) and nano- phytoplankton (2–20 µm) [27], as well as cyanobacteria [28,29]. However, this patter is less evident for viruses, bacteria, and zooplankton, as both increases or decreases in abundance have been reported under warming conditions. Using the same experimental design as the present study, Soulié *et al.* [30] and Zervouldaki *et al.* [31] documented shifts in plankton community functioning and metazooplankton communities in response to MHW. They reported that key functional processes, including phytoplankton growth (µ), and loss rate (l), were positively affected by the simulated + 3 °C MHW in phytoplankton functional groups such as diatoms, prymnesiophytes, and cyanobacteria, while dinoflagellates biomass declined [30]. Furthermore, the µ:l ratio shifted to negative over time, suggesting increased phytoplankton loss through grazing and/or viral lysis under warming [30]. Zervouldaki *et al.* [31] reported the dominance of meroplankton and harpacticoid copepods under MHW conditions, along with an increase in copepod offspring following the MHW [31]. To explore the response of a coastal Mediterranean planktonic community to a MHW, an in situ mesocosm experiment was conducted in a coastal lagoon (Thau Lagoon, Mediterranean Sea). A+3 °C increase above natural water temperature was simulated for ten days, during which the plankton communities (viruses, bacteria, phytoplankton, and protozooplankton) and environmental variables (temperature, salinity, light, nutrients, and chlorophyll-a) were recorded, followed by a ten-day post-stress period to study their resilience and recovery. This temperature increase is consistent with extreme MHWs frequently observed in the coastal Mediterranean Sea, which typically range from 2.8 to 3.6 °C [32]. Similarly, the duration of coastal MHW typically ranges from a few days to several weeks [5,32,33]; thus, a 10-day heat wave followed by 10 days of observation provides a suitable timeframe to highlight responses in the resilience and recovery of the community. By focusing on the microbial responses of marine plankton across multiple trophic levels (viruses, bacteria, phytoplankton, and protozooplankton), this study aimed to provide a more comprehensive understanding of how each plankton food web component and its interactions respond to MHW. In addition, we examined the compositional stability of the marine plankton communities by assessing their resistance, resilience, recovery, and temporal stability.

## Materials and methods

### Study site and experimental set up

The *in situ* mesocosm experiment was conducted in Thau Lagoon (Mediterranean Sea, south of France) at the Mediterranean Center for Marine Ecosystem Experimental Research (MEDIMEER) from May 24th to June 12th, 2019.

Thau Lagoon (75 km2), one of the largest Mediterranean coastal lagoons, supports major economic activities, including shellfish farming, tourism, and recreational activities [34]. The lagoon exhibits pronounced seasonal variability in temperature (3.7–29.6 °C) and phytoplankton biomass [35]. In the Thau lagoon, the mean annual temperature has increased from 15.3 °C to 16.8 °C (~ 1.5 °C) since 1998 [35]. A compilation of the available temperature data from Thau Lagoon revealed 36 occurrences of marine heat waves between 2011 and 2023. These events ranged in intensity from 2.7 to 5 °C and lasted from 5 to 25 days (S1 Fig).

Six mesocosms were directly immersed in the Thau lagoon around the MEDIMEER floating pontoon (43°24'53"N, 3°41'16"E). Each mesocosm was a 120 cm wide and 280 cm high transparent bag equipped with a 50 cm long conical sediment trap. The bags were made from 200 µm-thick vinyl acetate polyethylene film, which was reinforced with nylon (*Insinööritoimisto Haikonen Ky, Sipoo,* Finlande). To restrict inputs from precipitation and other external factors, the mesocosms were covered with polyvinyl chloride domes. Natural lagoon subsurface water was drawn using a pump (Rule, Model 360), filtered through a 1000 µm mesh to remove large particles and organisms, and collected in a container before being simultaneously distributed by gravity to all mesocosms via six parallel pipes. Each mesocosm was filled with 2200

litres, and a Rule model 360 pump was submerged at a 1 m depth in each mesocosm to ensure constantly mixing of the mesocosm water column (turnover rate: 3.5 d$^{-1}$) [30].

Over a 20-day experiment, two conditions were applied, each with three replicates. A MHW of 3 °C above the natural lagoon water temperature was simulated using a heating element (Galvatec, see details hereafter) and maintained in triplicate mesocosms from day 1 after water sampling to day 10 of the experiment, then the heating was stopped, and water temperature returned to the natural level of the lagoon for the next 10 days (heated condition 'MHW'). Three other mesocosms remained at the lagoon's natural temperature throughout 20 days of the experiment (control condition 'C'). Hereafter, MHW (heatwave) refers to the simulated heatwave period during the first 10 d, and PostMHW (post marine heatwave) refers to the last 10 d of the experiment after the heating was switched off (days 11–20). The MHW and PostMHW periods are also indicated in the control for comparison, although there was no heating in the control throughout the experiment. The submersible heating element was immersed to a depth of 1 m and automatically adjusted to maintain the temperature at + 3 °C in the MHW condition and compared to the control. This controlled system allowed the water column of the MHW condition to follow the natural daily fluctuations of the lagoon water [36,37] and ensured a consistent 3 °C difference between the MHW and control conditions during the 10 d of MHW period.

### High-frequency measurements and daily sampling

Water temperature was recorded using three temperature probes (*Campbell Scientific Thermistore Probe 107*) at three different depths (0.5, 1, and 1.5 m) at a high frequency (every 1 min). Additionally, as described in detail by Soulié et al. [30], a set of automated high-frequency sensors was installed in each mesocosm at a depth of 1 m to measure salinity, oxygen, photosynthetically active radiation (PAR), and fluorescence of chlorophyll-*a* every minute throughout the experiment. In this study, only temperature, salinity, and PAR sensors were used.

Nutrient concentrations, chlorophyll-*a*, viruses, bacteria, phytoplankton, and protozooplankton were measured once or twice a day at a depth of 1 m using a 5-litre Niskin water sampler.

### Analysis of abiotic variables

**Temperature, salinity and daily light measurements and treatment.** The temperature probe and conductivity sensors were calibrated before and after the experiment (refer to Soulié et al. [30] for details). Temperature and salinity data obtained at one-minute intervals were averaged daily. High-frequency measurements of PAR were used to determine the daily light integral (DLI), which represents the average amount of light received by a 1 m$^2$ surface over a 24-hour period [38], and expressed in mol·m$^{-2}$·d$^{-1}$.

**Nutrients and chlorophyll-*a* concentration analysis.** For the analysis of dissolved nutrients, daily water subsamples (50 mL) were collected in a Niskin bottle at a depth of 1 m and placed in an acid-washed polycarbonate bottle. Then, water samples were filtered through 0.45 µm filters (*Gelman*) in a polyethylene tube and stored at − 20 °C until analysis. The concentrations of nitrate [NO$_3^-$], nitrite [NO$_2^-$], ammonium [NH$_4^+$], orthophosphate [PO$_4^{3-}$], and silicate [SiO$_2$] were measured using an automated colorimeter (*Skalar Analytical, Breda,* The Netherlands). For the chlorophyll-*a* (chl-*a*) analyses, sub-samples (between 800−1200 mL) were taken daily from the Niskin bottle using acid-washed polycarbonate bottles, then water was filtered using Whatman glass-fiber filters, which were frozen in liquid nitrogen and stored at − 80 °C until analysis. Chl-*a* concentrations were measured using high-performance liquid chromatography (HPLC, Waters) according to Zapata et al. [39] and Vidussi et al. [37].

### Analysis of plankton communities

**Viruses, bacteria, pico- and nanophytoplankton analysis.** Characterization and counting of virus-like (hereafter referred to as virus), heterotrophic bacteria (thereafter simply called bacteria), and small phytoplankton, including prokaryotic and eukaryotic picophytoplankton (with cytometric estimated diameter around 0.2–3 µm), as well as

nanophytoplankton (with cytometric estimated diameter around 3–10 µm), were performed daily using different flow cytometers (see details hereafter). These instruments provide information regarding the approximate cell size, granularity, and fluorescence intensity through laser diffraction. Three 1.5 mL samples were fixed with glutaraldehyde Grade I (final concentration: 1.9% for viruses; 3.8% for bacteria and small phytoplankton) and preserved at – 80 °C until analysis.

Viral and bacterial samples were stained with SYBR Green I (final concentration of 0.25%) and analysed for three minutes at low flow rate with a FACS Canto II flow cytometer (Becton-Dickinson Biosciences, San Jose, CA, USA), using cytometry beads. Enumeration was based on side scatter and green fluorescence (530 nm) [40,41].

Phytoplanktons were analyzed using a CytoFLEX flow cytometer (Beckman Coulter, Miami, FL, USA) at a high flow rate for three minutes, also with cytometry beads. The phytoplankton groups were quantified based on orange (542–585 nm) and red (650 nm) autofluorescence and side and forward scatter signals.

One group of virus-like particles, one bacterium, and four phytoplankton species were identified. Cyanobacteria (hereafter referred to as Cyano, < 1 µm), grouping *Synechococcus*-like and *Prochlorococcus*-like (the latter being scarce), represent the picophytoplankton prokaryotes. Eukaryotic phytoplankton included picophytoplankton (Picophyto, 1–3 µm) and nanophytoplankton (Nanophyto, 3–10 µm).

**Large phytoplankton, heterotrophic flagellates, and ciliates analysis.** Abundance and diversity of larger phytoplankton (approximately 5–200 µm), heterotrophic or mixotrophic flagellates and dinoflagellates, and ciliates were assessed every two days (1, 3, 5, 7, 9, 11, 13, 15, 17, 19) using light microscopy (hereafter). Water samples were collected using a Niskin bottle at a depth of 1 m, placed in 100 mL dark glass bottles containing 3 mL of calcium carbonate-neutralized formaldehyde, and stored at 4 °C until analysis. Analyses were conducted at the Stazione Zoologica Anton Dohrn in Naples, following the sedimentation of 3 mL subsamples for 24 h in Utermöhl chambers. Cell counts and identification were performed using an inverted microscope (*Zeiss Axiovert 200*) at 400x magnification to determine the lowest possible taxonomic level. Phytoplankton cells above 5 µm were categorized into three groups: diatoms, dinoflagellates, and other flagellates. Among dinoflagellates and other flagellates, autotrophic, mixotrophic, and heterotrophic species were annotated and grouped according to their feeding strategies, based on information from the literature (see S1 Table). Organisms that could not be identified at the species or genus level and were classified into higher taxonomic groups (e.g., undetermined dinoflagellates = class level), and therefore potentially included various feeding strategies, were considered separately.

For heterotrophic nanoflagellates (HNF), 30 mL aliquots were also sampled using Niskin bottles at 1 m depth and preserved with formaldehyde (4% final concentration) at 4 °C in the dark until analysis. For HNF analysis, 10 mL subsamples were stained with DAPI (4,6-diamidino-2-phenyindole hydrochloride) and filtered onto 25 mm black nucleopore polycarbonate membranes (pore size 0.2 µm), which were placed on a white mixed cellulose ester membrane with a 3.0 µm pore size to ensure a more homogenous distribution of HNF on the filters. Each nucleopore polycarbonate membrane was then placed between coverslip and a slide with immersible oil, and HNF were counted using an epifluorescence microscope (Olympus AX70) at 100x magnification and classified into size groups of less or egal to 3, 3–5, and 5–10 µm (named thereafter HNF < 3 µm, HNF 3–5 µm, and HNF > 5 µm, respectively).

To identify and quantify aloricate ciliates and tintinnids, 125 mL aliquots of samples taken in Niskin bottles at 1 m depth were preserved in acid Lugol's solution (0.4% final concentration) and maintained at 4 °C in the dark until analysis. Samples (100 mL) were sedimented for 24 h in Utermöhl chambers in the dark and analyzed under an inverted microscope at 400x (Olympus IX-70).

The complete plankton community dataset is openly accessible through SEANOE at DOI:10.17882/108308.

## Calculations of resistance, resilience, recovery, and temporal stability

Resistance, resilience, recovery, and temporal stability were calculated according to the methods described by Hillebrand et al. [24] for viruses, bacteria, cyanobacteria, diatoms, dinoflagellates, flagellates for auto-, hetero- and mixotrophic,

undetermined groups, HNF, and ciliates. Additionally, a distinct group of very small algal cells, referred to as picophyto-plankton and nanophytoplankton were separated.

Community resistance, or the ability to withstand disturbance, was calculated for each day using available abundance data over the 10-day disturbance period, and then averaged. These values range from 0 to 1, representing low and high resistance, respectively.

Resilience, which reflects the ability of a species to recover after a disturbance, was calculated as the regression slope of community similarity over time (resilience = slope × time + intercept). A slope of 0 indicated no change in community similarity between conditions over the 10 days following disturbance. A positive slope suggests that the community tends towards a similar composition across conditions, indicating greater resilience. Conversely, a negative slope implied that communities within the conditions remained distinct, with their abundances diverging further from the control. This holds true when the effect of MHW on similarity is negative. The opposite is true when the effect is positive.

Recovery, defined as the ability of the observed variables to return to their initial conditions by the end of the experiment, was assessed on a scale of 0–1 and measured as the similarity value between conditions on the final day. A value of 1 represents a full recovery, whereas a value below 1 indicates a partial recovery.

Temporal stability examines the fluctuations in resilience during the recovery phase and is calculated as the inverse of the standard deviation of the resilience residuals. Higher values indicate smaller fluctuations around the recovery trend.

### Statistical analysis

Statistical analyses were performed using the R software (R Studio version 4.1.1). Repeated-measures analysis of variance (RM-ANOVA) was performed with treatment as a fixed effect and day as a random effect. When the assumptions for parametric tests, specifically normality of data and equality of variances, were not met, Kruskal-Wallis non-parametric tests were conducted. These analyses were carried out using packages *rstatix* version 0.7.2 and *ggpubr* version 0.5.0 [42,43]. Statistical tests were performed over three periods: (1) the marine heatwave (MHW) period, from day 2 when the temperature first reached + 3 °C, through day 10; (2) the PostMHW period, from day 11 to day 20; and (3) the entire experimental duration (days 2–20). In addition, the Student's t-test was applied to the compositional stability parameters to assess the significance of their differences from the benchmark associated with each parameter. Finally, the log ratio ($\log(\frac{mean\ value\ in\ MHW\ condition}{mean\ value\ in\ control})$) of the data was calculated and used to identify the relationships between the responses of various biological communities and abiotic variables. This was achieved using Principal Component Analysis ('PCA') using the *Factoextra* packages version 1.0.7 [44], which provides a simplified representation of the data while preserving most of the variance, thereby facilitating the interpretation of the correlations among these variables.

## Results

### Physical and chemical variables and chlorophyll-*a* response to warming

The temperature dynamics revealed a difference of approximately + 3 °C in the MHW condition relative to the control (C) during the MHW period, and similar water temperatures in both conditions during the PostMHW period (Fig 1A), after the heating was turned off. Temperature (°C), which reflects the natural temperature of the lagoon, fluctuated between 17.85 °C and 20.30 °C. For the MHW condition, the temperature ranged from 20.47 °C to 21.91 °C during the MHW period. Meanwhile, the temporal dynamics of salinity and daily light integral (DLI, Figs 1B and 1C) showed similar patterns in both conditions, with statistically significantly lower values in the MHW condition compared that of DLI in C (Table 1).

Orange and blue circles correspond to heated and control conditions, respectively. The gray area corresponds to the + 3 °C heating phase. Error bars indicate standard deviation of the mean.

Only the variables with at least one significant p-value (≤ 0.05) across the three periods are shown in bold, indicating a significant difference between the two conditions.

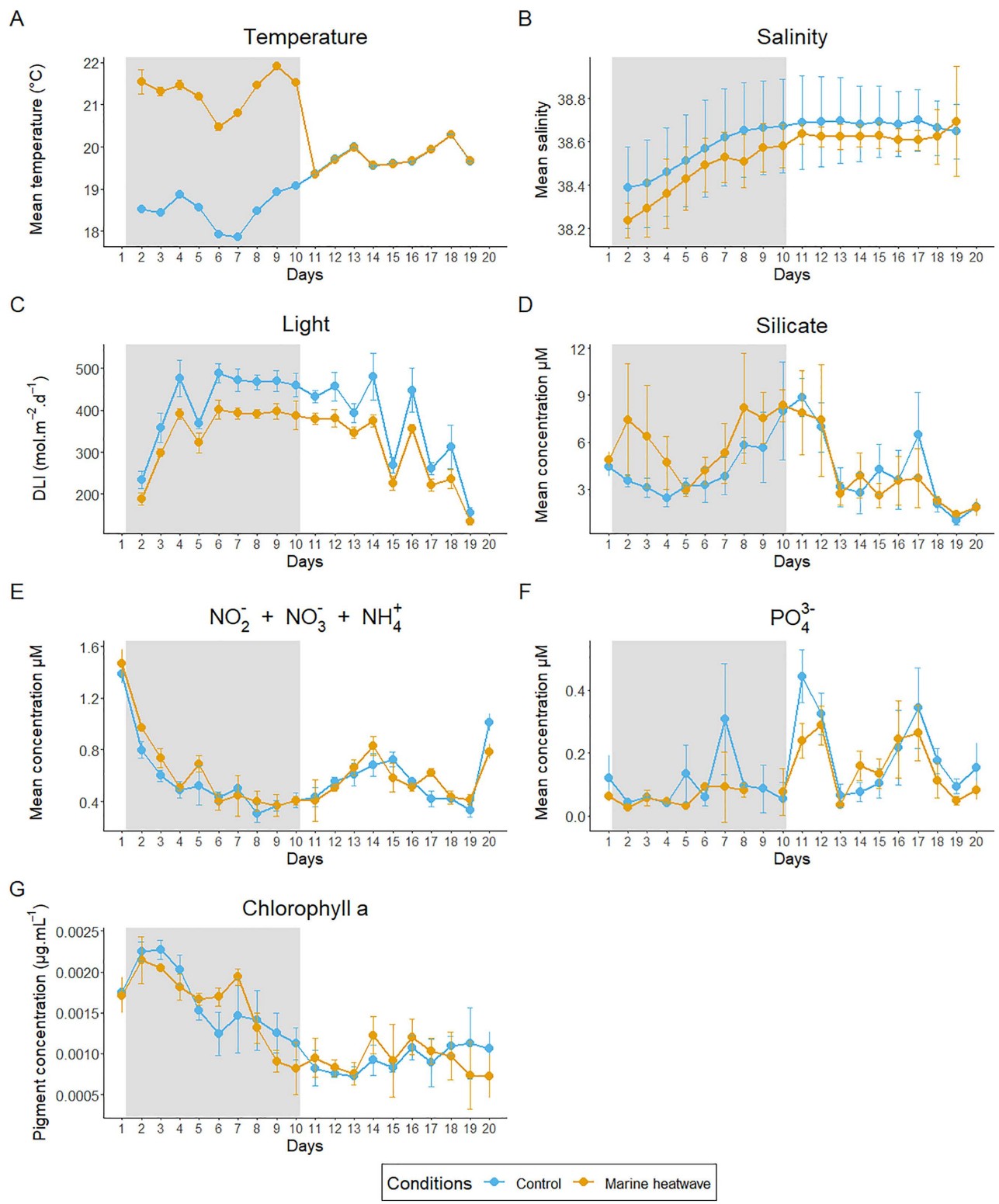

**Fig 1. Temporal mean changes physical and chemical variables, and chlorophyll a.** Physical variables; (A) Temperature, (B) salinity, and (C) Daily Light Integral ('Light'). Chemical concentrations; (D) silicate (Si), (E) nitrite + nitrate + ammonium ($NO_2^- + NO_3^- + NH_4^+$), (F) orthophosphate ($PO_4^{3-}$). (G) Chlorophyll A pigment.

**Table 1. P-value result, comparing conditions in abiotic and biotic variables across MHW, PostMHW and full experiment.**

| Parameters | MHW (d2-10) | | | PostMHW (d11-20) | | | ALL (d2-20) | | |
|---|---|---|---|---|---|---|---|---|---|
| | df | Test value | p-value | df | Test value | p-value | df | Test value | p-value |
| Temperature | 1 | **39.76** | **< 0.001 (a)** | 1 | 0.01 | 0.910 (a) | 1 | **44.61** | **< 0.001 (a)** |
| Daily light integral (DLI) | 44 | **123.84** | **< 0.001 (b)** | 1 | **7.42** | **0.006 (a)** | 1 | **16.15** | **< 0.001 (a)** |
| Salinity | 1 | 2.96 | 0.184 (a) | 1 | 0.13 | 0.723 (a) | 1 | 2.27 | 0.132 (a) |
| Si | 44 | **12.48** | **< 0.001 (b)** | 44 | 0.88 | 0.354 (b) | 1 | 1.02 | 0.312 (a) |
| $NO_3^- + NO_2^- + NH_4^+$ | 44 | **6.32** | **0.016 (b)** | 49 | $0.4E^{-2}$ | 0.239 (b) | 94 | 2.71 | 0.353 (b) |
| $PO_4^{3-}$ | 1 | 2.67 | 0.451 (a) | 49 | **4.17** | **0.046 (b)** | 1 | 2.06 | 0.378 (a) |
| Chlorophyll A | 44 | 0.15 | 0.706 (b) | 49 | $0.5E^{-2}$ | 0.526 (b) | 94 | 0.05 | 0.827 (b) |
| Viruses | 44 | **5.60** | **0.002 (b)** | 49 | 0.89 | 0.468 (b) | 1 | 0.30 | 0.309 (a) |
| Bacteria | 44 | 0.45 | 0.506 (b) | 49 | 0.08 | 0.953 (b) | 94 | 0.01 | 0.765 (b) |
| Cyanobacteria | 1 | 1.01 | 0.320 (a) | 1 | **8.31** | **0.003 (a)** | 1 | **5.76** | **0.027 (a)** |
| Picophyto | 44 | **31.88** | **< 0.001 (b)** | 1 | 0.80 | 0.154 (a) | 1 | 0.63 | 0.378 (a) |
| Nanophyto | 44 | 0.37 | 0.545 (b) | 49 | 0.04 | 0.841 (b) | 1 | $0.5E^{-3}$ | 0.975 (a) |
| Autotrophic flagellates | 1 | **4.49** | **0.032 (a)** | 1 | $0.8E^{-3}$ | 0.976 (a) | 1 | 2.23 | 0.135 (a) |
| Autotrophic dinoflagellates | 1 | 0 | 1 (a) | 1 | 3.51 | 0.131 (a) | 1 | 1.71 | 0.305 (a) |
| Diatoms | 1 | 0.33 | 0.564 (a) | 19 | **7.26** | **0.014 (b)** | 1 | 0.38 | 0.557 (a) |
| Mixotrophic flagellates | 19 | **0.36** | **0.002 (b)** | 19 | **5.08** | **0.036 (b)** | 39 | **17.07** | **< 0.001 (b)** |
| Mixotrophic dinoflagellates | 1 | 0.86 | 0.351 (a) | 1 | 0.18 | 0.668 (a) | 1 | 0.90 | 0.342 (a) |
| Heterotrophic nanoflagellates | 19 | **17.63** | **< 0.001 (b)** | 24 | 0.08 | 0.780 (b) | 44 | **9.90** | **0.003 (b)** |
| Heterotrophic dinoflagellates | 1 | 1.15 | 0.284 (a) | 1 | 0.22 | 0.641 (a) | 1 | 0.13 | 0.722 (a) |
| Undetermined dinoflagellates | 19 | 2.91 | 0.104 (b) | 19 | 2.44 | 0.135 (b) | 39 | $0.3E^{-4}$ | 0.995 (b) |
| Aloricate ciliates | 18 | **5.42** | **0.032 (b)** | 17 | 3.50 | 0.079 (b) | 36 | **7.33** | **0.010 (b)** |
| Tintinnids | 1 | 0.14 | 0.712 (a) | 1 | 0.75 | 0.387 (a) | 1 | 0.75 | 0.386 (a) |

(a) Kruskal-Wallis test, (b) repeated-measures ANOVA.

Silicate (Si) concentrations were relatively high (from 1 to 8.8 µM: Fig 1D), and were significantly higher in the MHW condition compared to the C during the MHW period (Fig 1D; Table 1), while

the dynamics were similar under both conditions. The concentration of N ($NO_2^- + NO_3^- + NH_4^+$) was also significantly higher in MHW than in C throughout the MHW period (Table 1). However, the dynamic was similar in both conditions, showing an initial decrease during the MHW period (from 1.46 to 0.31 µM) followed by a fluctuation in the PostMHW period, wherein concentrations remained low (less than 0.83 µM) until the end of the experiment (Fig 1E). Orthophosphate ($PO_4^{3-}$), concentrations were on average low (from 0.02 to 0.44 µM) and showed two peaks only in the C during the MHW period, while two additional peaks were observed during the PostMHW in both conditions (Fig 1F).

Chl-*a* concentrations at first increased reaching a maximum around $2.3 \times 10^{-3}$ µg·mL⁻¹ at day 2 in the MHW condition and the day after in the C before decreasing in both conditions, except a slight peak in the MHW condition in day 7, then reaching minimum concentrations around $0.7 \times 10^{-3}$ µg·mL⁻¹ in day 13 (Fig 1G). The chl-*a* concentrations fluctuated until the end of the experiment. No significant differences were observed between conditions.

## Plankton community responses to MHW

**Responses of viruses, bacteria, phytoplankton, mixo- and heterotrophic groups. Community dynamics.**
The abundances of bacteria, HNF, and diatoms progressively decreased from the start to the end of the experiment. Similarly, tintinnid and both mixotrophic flagellate and dinoflagellate abundances decreased after a few days. Bacterial

abundances ranged from $2.5 \times 10^6$ to $6.9 \times 10^6$ cells·mL$^{-1}$ and declined steadily in both conditions (Fig 2A). Diatoms and HNF peaked at the beginning, declined until day 11, and remained stable under both conditions thereafter (Figs 2B and 2C). Their abundances ranged from $0.5 \times 10^2$ to $110 \times 10^2$ cells·mL$^{-1}$ and $8.3 \times 10^2$ to $13.6 \times 10^2$ cells·mL$^{-1}$, respectively. Mixotrophic flagellates peaked on day 5 ($2.3 \times 10^3$ in HW; $3.5 \times 10^3$ cells·mL$^{-1}$ in C), followed by a gradual decrease (Fig 2D). Mixotrophic dinoflagellates abundances remained low (< 30 cells·mL$^{-1}$; Fig 2F) throughout the experiment. Tintinnids initially increased before rapidly declining under both conditions (< 0.3 cells·mL$^{-1}$), with the exception of a peak on the last day of MHW condition (Fig 2E).

Orange and blue circles correspond to heated and control conditions, respectively. The gray area corresponds to the + 3 °C heating phase. Error bars indicate standard deviation of the mean.

Viruses and undetermined dinoflagellates increased from the start of the experiment, peaking at the beginning of the PostMHW period, and remained stable thereafter, with abundance ranging from $3.7 \times 10^7$ to $6.8 \times 10^7$ cells·mL$^{-1}$ and from 41 to 303 cells·mL$^{-1}$, respectively (Figs 2G and 2H). Cyano, Picophyto, Nanophyto, and aloricate ciliates increased slightly after a period of decline or stabilization during the MHW period. Cyano and Picophyto underwent a decrease before increasing in both conditions, more rapidly in the MHW condition than in the C, with their abundance varying from $28.2 \times 10^3$ to $34.2 \times 10^3$ cells·mL$^{-1}$ and from $7.1 \times 10^3$ to $9.1 \times 10^3$ cells·mL$^{-1}$, respectively (Figs 2I and 2J). The Nanophyto peaking early (day 6), $10.7 \times 10^3$ cells·mL$^{-1}$, followed by a decline by day 13 and increased again to a maximum on day 17 under both conditions (Fig 2K). These cyanobacteria and nanophytoplankton dynamics indicate a bloom during the Post-MHW period. The aloricate ciliates abundance, ranging from 0.4 to 7.3 cells·mL$^{-1}$, decreased until day 9, peaking on day 11, then fluctuated downward for the remaining days of the experiment in both conditions (Fig 2L).

The abundance of autotrophic flagellates gradually increased only in the MHW condition, peaking on day 9 ($8.1 \times 10^2$ cells·mL$^{-1}$), and declined during the PostMHW period, while it remained consistently low in the control (< $1.7 \times 10^2$ cells·mL$^{-1}$; Fig 2M).

The autotrophic and heterotrophic dinoflagellate abundances remained low (< 30 cells·mL$^{-1}$) throughout the experiment under both conditions, with no clear temporal trends (Figs 2N and 2O).

**Effect of the heatwave.** The simulated MHW significantly affected several plankton community components. Virus, Picophyto, HNF, and Aloricate ciliate abundances were significantly lower in the MHW condition than in the control (C) during the MHW period, as were mixotrophic flagellates over the entire experiment. In contrast, the abundance of autotrophic flagellates was significantly higher under MHW conditions than under C conditions during the MHW period, as was the abundance of cyanobacteria and diatoms during the PostMHW period (Table 2). Other groups, including bacteria, nanophytoplankton, dinoflagellates (all trophic modes), and tintinnids, showed no significant differences between the conditions.

However, some trends were observed: Undetermined dinoflagellates tended to be higher in MHW than in C during the MHW period, with a reversed trend during the PostMHW period. Picophyto showed a shift between the MHW and Post-MHW periods, with a trend towards higher abundance in the MHW condition than in the C condition during the PostMHW period, whereas the reverse was observed during the MHW period.

**Communities and species of diatoms, flagellates, and dinoflagellates, all trophic modes, responses to MHW.** Some identified species were attributed to autotrophs, mixotrophs, heterotrophs, and undetermined species, as detailed in the Materials and Methods section (S1 Table).

**Autotrophic species community composition and responses to MHW.** The diatom community was initially dominated by *Cylindrotheca closterium* (Ehrenberg) Reimann & J.C.Lewin, during the MHW period (80% of the total community), while large pennate diatoms (> 10 µm) became dominant during the PostMHW period in both conditions (42% of the diatom community). Other notable taxa included *Bacteriastrum parallelum* Sarno, Zingone & Marino, *Chaetoceros* spp., *Cyclotella* spp., and small pennate diatoms (< 10 µm), contributing between 11% and 34% of the diatom community.

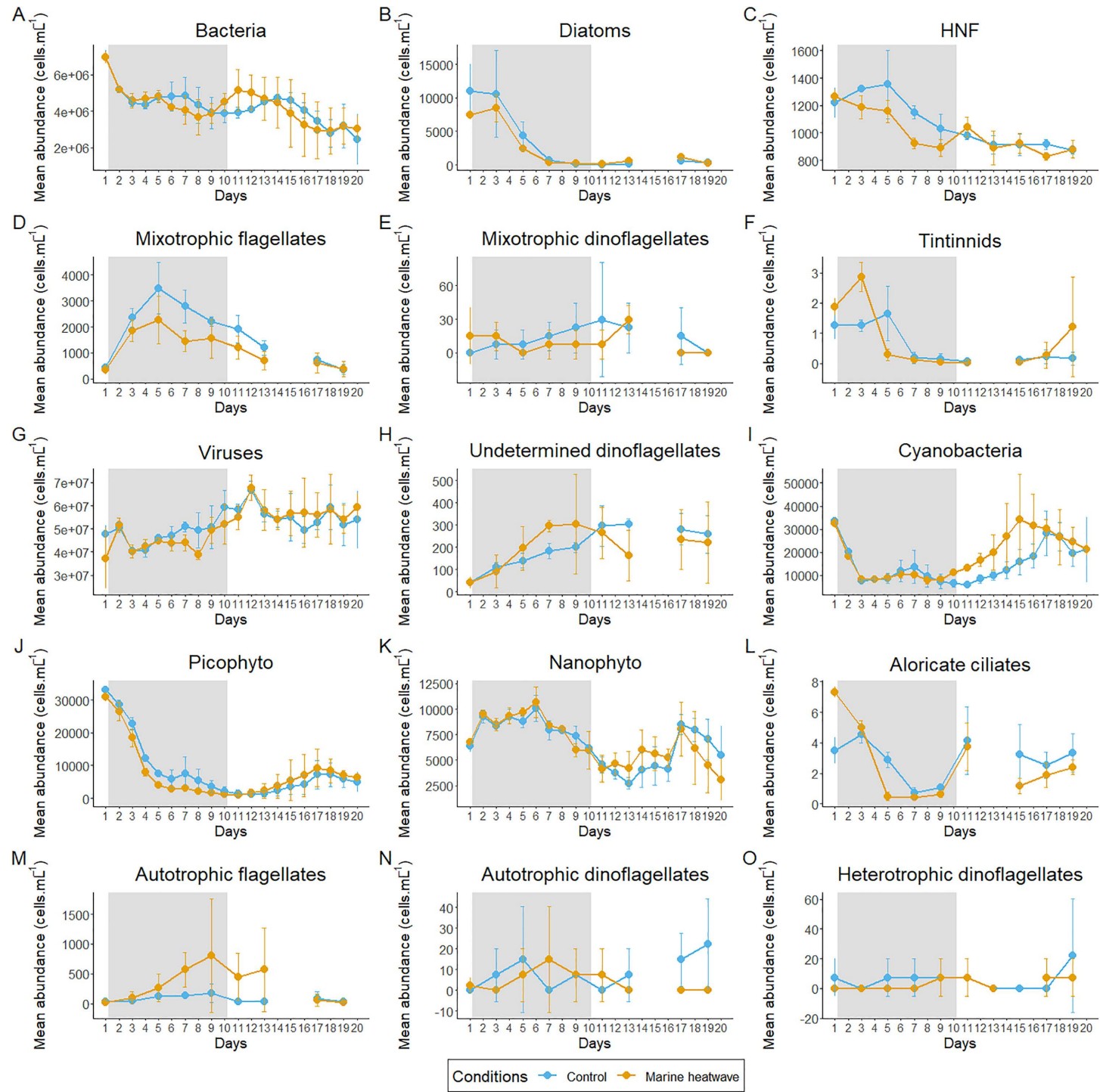

**Fig 2. Dynamic in daily and bi-daily average abundance of plankton community.** Cytometric populations; (A) bacteria, (G) viruses, (I) Cyanobacteria (Cyano), (J) Picophytoplankton eukaryotes (Picophyto), (K) Nanophytoplankton eukaryotes (Nanophyto). Microscopy community; (B) diatoms, autotrophic (M) flagellates and (N) dinoflagellates, mixotrophic (D) flagellates and (E) dinoflagellates, heterotrophic (C) nanoflagellates (HNF) and (O) dinoflagellates, (H) undetermined dinoflagellates, (L) aloricate ciliates, and (F) tintinnids.

**Table 2. Compositional stability of organisms, including average resistance, resilience, recovery and temporal stability.**

| Groups | Resistance | Resilience | Recovery | Temporal stability |
|---|---|---|---|---|
| Viruses | **0.96 ± 0.038** | −2.04E-03 | 0.95 | 50.42 |
| Bacteria | **0.96 ± 0.37** | 4.64E-03 | 0.90 | 22.73 |
| Cyanobacteria | **0.92 ± 0.082** | 4.72E-02 | 0.99 | 12.4 |
| Picophyto | **0.72 ± 0.135** | 1.13E-02 | 0.87 | 18.01 |
| Nanophyto | **0.97 ± 0.031** | −1.17E-02 | 0.72 | 13.81 |
| Diatoms | **0.77 ± 0.104** | 5.74E-02 | 0.84 | 8.97 |
| Autotrophic flagellates | **0.52 ± 0.168** | 7.79E-02 | 0.57 | 5.05 |
| Autotrophic dinoflagellates | 0.42 ± 0.5 | 0.00E+00 | 0.00 | Inf |
| Mixotrophic flagellates | **0.80 ± 0.085** | 2.85E-02 | 0.98 | 26.92 |
| Mixotrophic dinoflagellates | **0.46 ± 0.316** | −8.79E-02 | NA | 2.99 |
| Heterotrophic nanoflagellates | **0.92 ± 0.023** | 4.71E-04 | 1.00 | 50.28 |
| Heterotrophic dinoflagellates | 0.33 ± 0.517 | −8.65E-02 | 0.50 | 2.88 |
| Undetermined dinoflagellates | **0.83 ± 0.059** | 8.45E-03 | 0.92 | 8.98 |
| Aloricate ciliates | 0.68 ± 0.279 | −9.71E-03 | 0.84 | 5.68 |
| Tintinnids | **0.56 ± 0.191** | −2.36E-02 | 0.24 | 3.75 |

The dynamics of *C. closterium, B. parallelum*, and *Chaetoceros* spp. were similar across conditions and reflected the total diatom trends (Fig 2B; S2A-S2C Figs). No significant differences were observed between the conditions, although *B. parallelum* tended to be more abundant under MHW conditions than under C conditions during the PostMHW period.

Large and small pennate diatoms remained low overall (less than 88 cells·mL⁻¹ and 100 cells·mL⁻¹, respectively) but increased during the PostMHW period, with small pennates being significantly more abundant in the MHW condition than in C (S2 Table; S2D and S2E Figs). *Cyclotella* spp. were significantly more abundant in the C condition than in the MHW condition during the MHW period, although their overall abundance remained low (0–104 cells·mL⁻¹; S2F Fig). The patterns of *Pseudoscourfieldia marina* (Throndsen) Manton and *Prorocentrum triestinum* J. Schiller followed those of the total autotrophic flagellates and dinoflagellates, respectively (Figs 2M and 2N; S2G and S2H Fig).

**Mixotrophic, heterotrophic, and undetermined species composition and responses to HW.** The mixotrophic flagellates *Dinobryon faculiferum* Willén (Willén) and *Ollicola vangoorii* (W.Conrad) Vørs were present in both conditions (S1 Table). *O. vangoorii* showed a dynamic similar to that of the total mixotrophic flagellates and was significantly lower in the MHW condition than in the C condition (S2 Table; S2I Fig), whereas *D. faculiferum* showed no significant differences between the conditions (S2J Fig).

Among dinoflagellates, small (< 15 µm) thecate and naked taxa were the most abundant throughout the experiment. The naked dinoflagellates < 15 µm predominated in the MHW condition (69% during MHW period, 58% during the PostMHW period), with significantly higher abundance compared to C during the MHW period (S2 Table; S2K Fig). Conversely, thecate dinoflagellates < 15 µm were dominant in the C (42% in MHW period, 69% in PostMHW period), with significantly lower abundance in MHW condition compared to the C during the PostMHW period (S2 Table; S2L Fig).

Mixotrophic (*Prorocentrum gracile* F. Schütt) and heterotrophic dinoflagellates (*Gyrodinium* spp., *Protoperidinium bipes* (Paulsen) Balech, and *Protoperidinium diabolus* (Cleve) Balech) remained low (< 30 cells·mL⁻¹) under both conditions with no significant MHW effect (S2M and S2O Figs).

The majority of HNF were < 5 µm. HNF < 3 and HNF 3−5 µm ranged from 310 to 748 cells·mL⁻¹ (S3A and S3B Figs). The HNF < 3 abundance was significantly higher in the MHW condition than in C during the PostMHW period, while HNF 3−5 µm was a significantly lower across both periods (S2 Table). HNF > 5 µm remained scarce than in the other groups, ranging from 16 to 115 cells·mL⁻¹ (S3C Fig).

## Community stability

The mean resistance values of viruses, bacteria, cyanobacteria, nanophytoplankton, undetermined dinoflagellates, and HNF ranged from 0.83 ± 0.06 to 0.97 ± 0.03 (Table 2). Although relatively close to the reference value (1), suggesting a higher resistance during the MHW period, these values were significantly lower than the reference values (Fig. 3). In comparison, picophytoplankton, diatoms, flagellates (autotrophic and mixotrophic), aloricate ciliates, and tintinnids displayed lower mean resistance values ranging from 0.52 ± 0.17 and 0.80 ± 0.09. Only autotrophic, mixotrophic, and heterotrophic dinoflagellates exhibited values below 0.5, indicating potentially lower resistance to MHW (Table 2). All these values were significantly different from the reference values, except for those of aloricate ciliates and both autotrophic and heterotrophic dinoflagellates (Table 2). However, it is important to consider the variability in the resistance values and their associated standard deviations when interpreting the overall response of each group, as illustrated in the boxplot (Fig 3).

The significantly different between the values and the Resistance benchmark are indicated in Bold with p-value ≤ 0.05.

Resistance benchmark: 1 = maximum resistance and < 1 = low resistance. Resilience benchmark: 0 = no recovery, > 0 (faster) recovery, and < 0 deviation from control. Recovery benchmark: 1 = maximum recovery and < 1 incomplete recovery

The error bar indicates the standard deviation of the mean.

Overall, the community resilience values were close to the reference value (0), indicating low resilience. Cyanobacteria, picophytoplankton, diatoms, and autotrophic flagellates displayed the highest values among the groups studied, ranging from 0.0113 to 0.0779, suggesting divergence from the control owing to the positive effect of MHW on similarity. In contrast, mixotrophic flagellates exhibited a resilience value of 0.03 and a negative MHW effect on similarity, indicating good

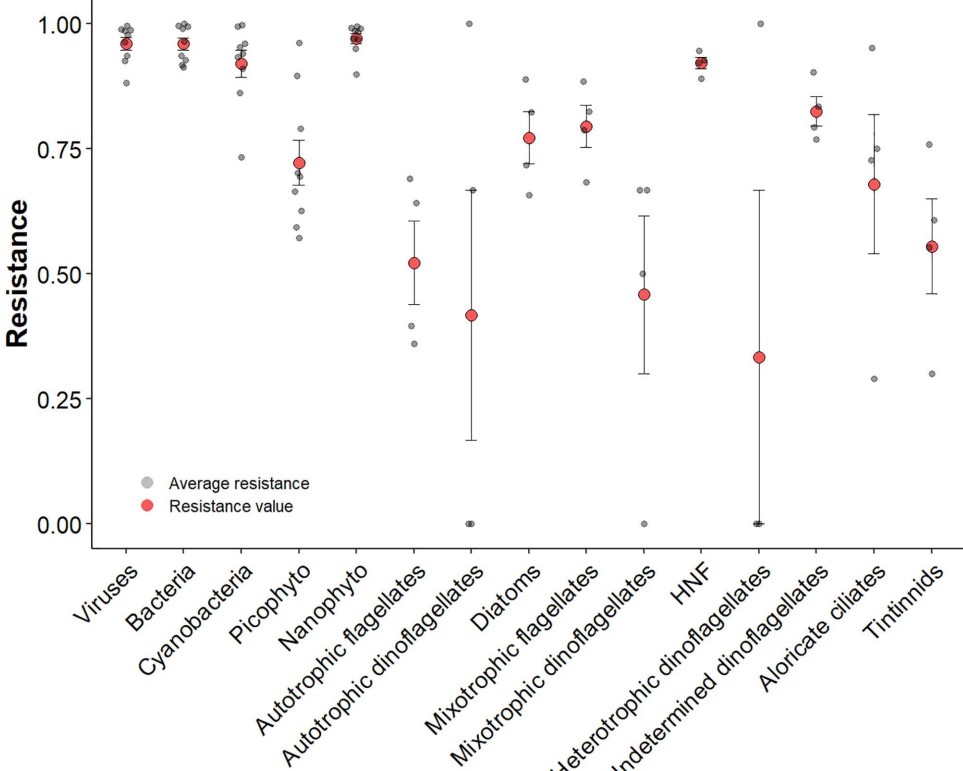

**Fig 3. Boxplot of the average resistance regarding MHW for different organisms which identified in the X axis.**

resilience during the PostMHW period. Regarding the ability of organisms to return to their "control state" at the end of the experiment (recovery), viruses, bacteria, cyano, picophyto, diatoms, HNF, mixotrophic flagellates, aloricate ciliates, and undetermined dinoflagellates almost or completely returned to the control state, with values ranging from 0.81 to 1 (Table 2, Recovery). Nanophytoplankton, autotrophic flagellates, and heterotrophic dinoflagellates exhibited partial recovery, reaching approximately half of their initial state. Only tintinnids did not return to their "Initial state" as indicated by the value of 0.24. Finally, the temporal stabilities of viruses, bacteria, and HNF ranging from 22.73 to 50.42 were higher than those of diatoms, autotrophic flagellates, dinoflagellates (mixotrophic, heterotrophic and undetermined), aloricate ciliates, and tintinnids, which ranged from 2.88 to 8.98. The resilience, recovery, and temporal stability of autotrophic dinoflagellates were not calculated because they were not observed during the PostMHW period.

## Relationships between biological communities and the abiotic variables

The PCA(Fig. 4) show only variables contributing up to 5% of the axis. The first two dimensions of the PCA together accounted for 54.2% in MHW period and 56.4% in PostMHW period of total variance in the dataset.

The first dimension in the MHW period and the second in the PostMHW period were positively represented by the bacterial and HNF communities, indicating a positive relationship between these two groups regardless of the treatment period (Figs 4A and 4B). During the MHW period, the virus was also on the positive scale of the first axis with bacteria and HNF, which was not the case during PostMHW, suggesting a virus-bacteria and virus-HNF mismatch during

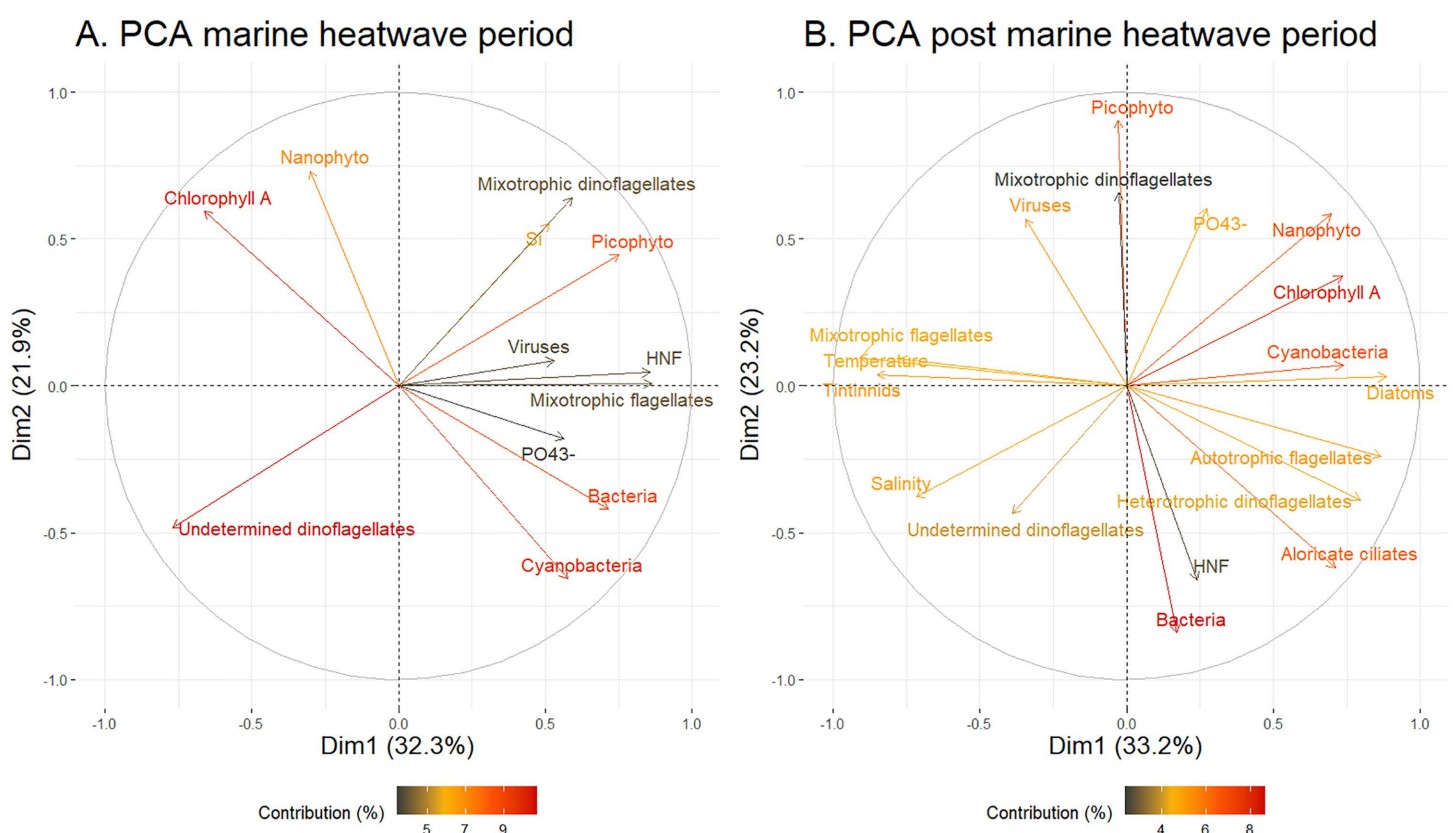

**Fig 4. PCA analysis of the log ratio responses of environmental variables and biological communities.** (A) MHW period and (B) PostMHW period. Only variables with contribution above 5% to the dimensions 1 or 2 were shown in the figure. The contribution is express in percentage (%).

PostMHW (Figs 4A and 4B). Picophytoplankton and mixotrophic flagellates were also part of this group but only during the MHW period, as they were either opposed to or uncorrelated with bacteria and HNF during the PostMHW period. The Nanophyto community was most closely related to the concentrations of chl-a in both periods, suggesting that Nanophyto was the major contributor to phytoplankton chl-a biomass in the present study. Both groups were only related to cyanobacteria-, diatom-, and autotrophic dinoflagellates during the PostMHW period (Figs 4A and 4B). Additionally, Cyano and diatoms were located on the positive scale of the first dimension, whereas mixotrophic flagellates, tintinnids, and temperature were positioned at the opposite end of the axis during the PostMHW period.

## Discussion

This *in situ* mesocosm study aimed to understand how plankton communities were affected by a simulated marine heat wave (MHW) in the Thau Lagoon. The first primary finding highlighted that the responses to MHW at the community level differed according to the food strategies of the microorganisms. Protozooplankton responded negatively to warming throughout the experiment, whereas autotrophs and undetermined microorganisms responded positively to MHW during both MHW and PostMHW periods. The second primary observation was that the MHW was beneficial for smaller organisms, mainly those <10 μm, whatever their feeding strategy, which exhibited the highest resistance throughout the MHW period.

### Asymmetric responses of the plankton community to marine heatwave

MHW significantly reduced the abundance of protozooplankton (nanoflagellates, aloricate ciliates, tintinnids, mixotrophic flagellates, and dinoflagellates) during the 20-day experiment. This significant decline, especially in aloricate ciliates and tintinnids, aligns with previous studies that showed the sensitivity of these groups to thermal conditions [37,45]. The reduction in heterotrophic nanoflagellates (HNF) by MHW was consistent with earlier findings that reported a decrease in their abundance and biomass under warm conditions [45,46]. HNF, mixotrophic flagellates (*D. faculiferum* and *O. vangoorii*), and dinoflagellates (*P. gracile*) were also reduced following MHW. Some ciliate species and heterotrophic and mixotrophic dinoflagellates may reach the limits of their metabolic equilibrium as temperature rises beyond 19 °C [47]. This threshold was exceeded (21 °C) during the MHW period, which explains the general reduction in activity due to metabolic constraints. This contrasts with previous observations made in the same lagoon during early spring the previous year, where a more moderate temperature (not exceeding 19 °C) favored the proliferation of dinoflagellates and small green algae [48].

MHW had significant effects on phytoplankton abundance at both the community and temporal scales. In line with the observed pattern, the phytoplankton abundance, including that of cyanobacteria, pico- and nanophytoplankton, and diatoms, declined progressively under both conditions. This decline could be attributed to nutrient limitation and increased grazing by metazooplankton (copepods and polychaete larvae) and protozooplankton. Picophytoplankton and *Cyclotella* spp. decreased more rapidly under HW conditions, consistent with earlier observations from the Thau lagoon one year prior, where a +3 °C warming negatively impacted eukaryotic picophytoplankton [49]. This could be linked to the simultaneous increase in undetermined dinoflagellates, for which top-down control was strengthened by MHW. Additionally, warming may impair nitrate assimilation and uptake by diatoms [50,51], possibly affecting *Cyclotella* spp.

Interestingly, a delayed positive effect of MHW was observed for cyanobacteria and diatoms during the PostMHW period, with a similar trend observed for picophytoplankton. This suggests a relaxation of predation by ciliates, dinoflagellates, and HNF, whose abundances decreased. Such patterns support the view that MHW primarily exerts indirect effects on phytoplankton via microzooplankton decline [52–54]. *P. marina*, an autotrophic flagellate, was the only species that showed an immediate positive response to MHW, likely due to its thermal tolerance or reduced competition [55].

The MHW also exerted strong selective pressure in favor of smaller species, such as *B. parallelum, Chaetoceros* spp., and *P. marina* with sizes <15 μm [56,57], as well as picophytoplankton, consistent with studies showing that warming benefits small phytoplankton and cyanobacteria [27–29]. In contrast, larger diatoms like *C. closterium* (ranging from around 30–160 μm in length) and *Cyclotella* spp. (ranging from 8 to 80 μm in diameter) did not increase during the PostMHW

period, regardless of conditions. The predominance of small undetermined dinoflagellates (< 15 μm) under MHW further underscores the advantage of small cell size under warming conditions. The shift from large to small species at elevated temperatures has been well-documented [58–60], confirming that it is a generally observed rule, at least in temperate coastal waters, and may have important consequences for primary consumers [61].

The reduction in chl-*a* concentration mirrors these changes, indicating a drop in both larger phytoplankton groups and overall biomass, as chl-*a* correlates more closely with biomass than with abundance. PCA analyses highlighted a coupling between chl-*a* and nanophytoplankton contributing most to chl-*a* in both periods (Figs 4A and 4B), but their small size (3–10 μm) limited their biomass contribution despite rising abundance. These results align with global projections of declining chl-*a* levels under ocean warming [62].

Unlike protozooplankton and phytoplankton, bacteria did not exhibit a clear response to MHW. Such heterogeneous dynamics have been reported, including decreases during spring blooms in Thau Lagoon [37] and increases in autumn, winter, and spring blooms in coastal areas [28,29]. This variability likely reflects a high functional diversity, sensitivity to environmental conditions, and multiple interacting factors [61,63]. Despite their theoretically higher metabolic sensitivity to warming [49,64], bacterial responses may be constrained by nutrient limitation following phytoplankton uptake, competition with fast-growing autotrophs, or enhanced top-down control by viral lysis and protozooplankton grazing [28,30,65]. These results underscore the complexity of the microbial interactions and environmental drivers that shape bacterial dynamics during marine heatwaves.

The abundance of viruses increased over the 20-day experiment but remained significantly lower under heated condition during the MHW period. In contrast, some studies have reported a non-significant positive effect of warming on viruses [28,37], highlighting that viral responses likely depend on the structure and dynamics of co-occurring plankton communities under MHW. This interplay may influence microbial food web processes and shape the dynamics of the viral community.

Additionally, Zervouldaki et al. (2024) reported that the copepods, polychaetes, larvae of bivalve, and gastropods, were the dominant mesozooplankton groups present in the mesocosms during the same experiment The MHW condition had a significant positive effect on copepods, mainly driven by an increase in harpacticoid copepods, while a positive trend was also observed for polychaete larvae [31]. These results are consistent with previous studies demonstrating that warming can stimulate mesozooplankton development at the same location [37] and enhance zooplankton grazing in the Baltic sea [66]. Collectively, these findings highlight that the decrease in the abundances of protozooplankton and phytoplankton communities may be mediated by increased predation pressure exerted by mesozooplankton.

## Marine heatwave-induced trophic cascades restructure the planktonic food web

The aforementioned trophic interactions contributed to the initial phytoplankton decline. The rapid increase in HNF > 5 μm, combined with the small dinoflagellates present in heated mesocosms, may have contributed to the declines in cyanobacteria and picophytoplankton. Similarly, the high abundance of ciliates early in the experiment likely contributed to the initial diatom decline, which is consistent with studies showing increased protozooplankton predation and reproduction under warming conditions [67–69]. This underlines the strengthened top-down control of phytoplankton by protozooplankton at the onset of the experiment, leading to reduced cyanobacteria and picophytoplankton abundance. The first protozooplankton abundance peak was soon followed by increased metazooplankton, harpacticoid copepods, and polychaete larvae [31], suggesting that warming intensified the top-down control of both diatoms and protists. The Metabolic Theory of Ecology and several other studies support this hypothesis, showing that grazing rates increase more with temperature than with phytoplankton growth [67,68,70], amplifying plankton prey consumption. In addition, Soulié et al. (2022) reported that under MHW, the system tended to shift toward heterotrophy due to increased respiration and phytoplankton mortality [30], further supporting the idea that mesozooplankton exerted a strong top-down control on diatoms and protozooplankton under MHW condition. These outcomes could also impact biogeochemical cycles, particularly the oxygen cycle, where

oxygen consumption may exceed production under MHW, with cascading consequences for the carbon cycle through an intensified release of carbon into the water. Consequently, under MHW, coastal Mediterranean systems may act as a carbon source to the atmosphere rather than a sink of carbon. Moreover, silica accumulation upon heating suggests diatom mortality via silicate dissolution, which is consistent with a decrease in diatom abundance.

The subsequent decline in protozooplankton had major trophic repercussions. Simultaneously, the abundance of metazooplankton, particularly harpacticoid copepods and meroplankton (e.g., polychaete larvae), increased [31]. These taxa preferentially feed on dinoflagellates and ciliates rather than phytoplankton [14,31]. A shift in dominance from copepods to meroplankton may have intensified protozooplankton decline, possibly through resource competition exacerbated by warming, which increases grazing by protozooplankton and meroplankton larvae on small particles, such as phytoplankton and bacteria [37,45]. Late-stage larval predation on protists may have contributed further. Moreover, approaching thermal limits for some ciliates (~22 °C) during the MHW condition (20.4–21.9 °C) could have increased physiological mortality compared to their respective predators [47].

The decline in ciliates reduces grazing on nanoplanktons, which are their main prey [14]. However, HNF also declined during MHW period despite lower protozooplankton predation. This suggests other drivers, such as competition with undetermined dinoflagellates, resource availability and quality [61], and viral pressure [71–73]. Some studies have shown reduced top-down control by protozooplankton during MHWs [28], while others have reported intensified grazing [48] or phytoplankton losses driven by microzooplankton, depending on the trophic status of the water (e.g., eutrophic vs. oligotrophic system) [67]. Overall, the balance between protozooplankton predation, HNF, and undetermined dinoflagellate competition and the thermal tolerance of each community appears to be key to planktonic community structuring under marine heat waves. This highlights the importance of multitrophic interactions in shaping the planktonic responses to extreme warming.

The PostMHW resurgence of cyanobacteria, picophytoplankton, and diatoms suggests that reduced predation by metazoans and protozoans is a natural trophic succession. This delayed positive effect reinforces the idea that this response was more of an indirect effect of predation than a direct response to MHW, which was evident in protozooplankton. This suggests a long-term effect of elevated temperature and introduces the concepts of resistance and resilience. The substantial increase in metazooplankton offspring in the PostMHW period also supports reduced protozooplankton grazing on phytoplankton due to the marine heatwave-induced alteration of trophic cascades [31]. This shift likely resulted in the cascading release of phytoplankton groups, particularly cytometric populations, from predation pressure following MHW period [28,37,66]. Alternatively, meroplankton larvae may have switched from stationary (i.e., diatoms) to mobile prey (i.e., ciliates, HNF) to meet the rising energy demands under warming [74].

Finally, complex trophic pathways may favor cyanobacteria and picophytoplankton. Viral regulation of HNF may release dissolved organic matter (DOM), benefiting cyanobacterial growth and reducing HNF grazing pressure [75,76]. Viral lysis, a major source of microbial mortality [71], is expected to intensify with rising temperatures due to enhanced viral production [72], releasing DOM that is essential for cyanobacteria [77]. In support of this, high-frequency chl-*a* fluorescence data revealed a negative phytoplankton growth:loss ratio under MHW condition, indicating increased grazing and viral lysis [30]. Consequently, viruses may have contributed to the observed reductions in the abundances of bacteria, picophytoplankton, and HNF < 5 µm during the MHW period, while promoting cyanobacteria growth through potential mutualistic interactions, particularly during the PostMHW period. Meanwhile, the PostMHW period decline in dinoflagellate and meroplankton predation could alleviate competition, further aiding picoplankton proliferation.

These mechanisms highlight how MHW can indirectly reshape plankton communities by altering trophic interactions and potentially enhancing primary production by reducing protozooplankton grazing on phytoplankton.

## Resistance does not ensure resilience: long-term repercussion of a short marine heatwave

Smaller-cell-sized communities that showed the highest resistance values (autotrophic, mixotrophic, or heterotrophic), while those of larger communities were varied.

Competition intensified by MHW appears to manifest primarily through the greater resistance of smaller-cell-sized communities, which adapt or acclimate more rapidly. In contrast, larger-cell communities struggle to reach equilibrium under MHW stress owing to top-down and bottom-up controls. This aligns with the observation that the competitive advantage of smaller species is reinforced in disturbed environments [78]. Moreover, small microbial plankton (viruses, bacteria, cyanobacteria, nanophytoplankton, HNF, and undetermined dinoflagellates), with the highest resistance values (> 0.9), differed significantly from the reference value (1), whereas the large ones (aloricate ciliates, autotrophic, and heterotrophic dinoflagellates), with low values, showed no difference. The low resistance values of these last groups, as well as the significant variability around the mean, raise questions regarding the statistical validity of the test. According to Pimm (1984), resistance refers to the extent to which a community remains unaffected by disturbances [79]. Therefore, it is not possible to confirm that these communities are resistant due to the lack of statistical significance in the test compared to the benchmark. This could be due to the fact that these communities were sampled every two days, unlike the cytometric groups sampled every day. This reduces the number of observations and potentially introduces bias into statistical analyses because resistance is a stability property that is inherently challenging to quantify [80–82]. Therefore, future experiments should sample all organisms in the marine plankton food web at the same time frequency to limit potential biases from a lack of data. However, it is clear that this cannot be achieved with classical microscopic methods as they require an excessive effort of analysis.

Resilience and recovery were calculated to assess the ability of communities to return to their normal state after a disturbance. The resilience slope values were generally close to zero or negative, which is not surprising for resistant organisms, but is questionable for those that are not. Higher resistance reflects minimal variation between conditions, suggesting that the community remains stable and does not require resilience, as it is inherently resistant to disturbance. Cyanobacteria, picophytoplankton, diatom, autotrophic, and mixotrophic flagellates showed the highest positive resilience slopes compared to the other groups (viruses, bacteria, nanophyto, HNF, all trophic modes dinoflagellates, and ciliates) that exhibited negative or near-zero slopes. This result indicated the strong resilience of the mixotrophic flagellates, while the other groups diverged more from the control group in terms of the positive effect of MHW condition on similarity. This finding contradicts that of another study that showed the strong resilience of phytoplankton communities to MHW [60].

Additionally, at the end of the experiment, the planktonic community displayed varying degrees of resilience among the different groups. Nanophytoplankton, autotrophic flagellates, and heterotrophic dinoflagellates only partially recovered, whereas tintinnids exhibited minimal recovery. In contrast, viruses, bacteria, cyanobacteria, picophytoplanktons, diatoms, HNF, aloricate ciliates, mixotrophic flagellates, and both mixotrophic and undetermined dinoflagellates demonstrated a strong capacity for full or partial recovery following MHW period. This indicates that the planktonic microbial community has the ability to return to their initial state more rapidly, favoring their development and proliferation at the expense of organisms with slower or more difficult recovery periods [83]. This suggests that despite the initial resistance observed during the MHW period, long-term repercussions may arise following a marine heatwave. This may result from rapid, potentially irreversible acclimation, pushing communities towards a tipping point and trigerring lasting community shifts once conditions normalize. Thus, a moderate MHW of 3 °C lasting 10 days should not be seen as a transient event but as a significant disturbance that alters plankton communities and pushes them towards a new ecosystem state. The transition between these states represents a period of instability and vulnerability in planktonic communities. This increases the likelihood that these communities will evolve or shift towards an alternative stable state that is potentially distinct from that prior to disturbance [84].

Both bottom-up and top-down controls play a role in community stability. For example, some predators within a plankton community, although sufficiently resilient to survive the disturbance, may lack the ability to withstand it without experiencing negative effects. Once conditions stabilize, resumed predation can alter prey population dynamics, thereby reshaping community structure. This implies that plankton communities remain reliant on top-down effects mediated by the effect of temperature on predation [85,86]. Over the past 15 years, the onset of spring blooms in Oslofjorden has been delayed due to rising temperatures in winter and spring [87], similar to the pattern observed in diatoms, In contrast,

small green flagellates showed the opposite response under warming [48]. This could disrupt the planktonic food web by altering the timing of species and communit dynamics. Given the low likelihood that the frequency and intensity of MHWs or surface warming will decline [2,3,6], predicting the trajectory of planktonic communities and the ecosystem adaptation remains challenging [88]. In addition, shorter intervals between MHWs may prevent the full recovery of planktonic groups after each disturbance, weakening community resilience and potentially leading to long-term decline. This can alter interactions within the marine planktonic food web and disrupt overall ecosystem function.

Overall, this study shows that marine heatwaves can profoundly and indirectly reshape coastal planktonic food webs by altering trophic interactions, favoring smaller taxa, and destabilizing the balance between resistance and resilience. By capturing multitrophic dynamics over time, our findings provide key insights into the vulnerability and adaptive capacity of plankton communities facing increasingly frequent thermal disturbances.

## Supporting information

**S1 Table. Summary of identified organisms using microscopy and feeding strategies derived by literature.** All species identified across the different planktonic groups are listed to highlight overall richness. Some species were not discussed in the main text due to their very low abundance. Autotrophic (A); Mixotrophic (M); Heterotrophic (H); Undetermined (U).
(DOCX)

**S2 Table. P-value results, comparing conditions in plankton community species, genus, and group.** (a) Kruskal-Wallis test, (b) repeated-measures ANOVA. Only the variables with at least one significant p-value ($\leq 0.05$) across the three periods are shown in bold, indicating a significant difference between the two conditions.
(DOCX)

**S1 Fig. Number of marine heatwaves events as function of their intensity occurring in the Thau lagoon.** The data coming from ECOSCOPA network from 2011 to 2023.
(TIF)

**S2 Fig. Dynamic of the mean abundance of the species, genus and group of plankton community.** Diatoms communities: (A) *Cylindrotheca closterium*, (B) *Bacteriastrum parallelum*, (C) *Chaetoceros* spp, (D) the pennate diatoms with the size superior to 10 µm (pennate > 10 µm), and (E) with a size under 10 µm (pennate < 10 µm), (F) Cyclotella spp., autotrophic flagellates: (G) *Pseudoscourfielda marina*, autotrophic dinoflagellates; (H) *Prorocentrum triestinum*, mixotrophic flagellates: (I) *Ollicola vangoorii* and (J) *Dinobryon* faculciferum, undetermined dinoflagellates: (K) naked dinoflagellates < 15, (L) thecate dinoflagellates < 15, mixotrophic dinoflagellates: (M) *Prorocentrum gracile*, heterotrophic dinoflagellates: (N) *Gyrodinium spp*. Orange and blue circles correspond to heated and control conditions, respectively. The gray area corresponds to the + 3 °C heating phase. Error bars indicate standard deviation of the mean.
(TIF)

**S3 Fig. Dynamic of the mean abundance of heterotrophic nanoflagellates.** (A) with body size inferior to 3 µm, HNF < 3 µm, (B) between 3 µm and 5 µm, HNF 3–5 µm, and (C) superior to 5 µm, HNF > 5 µm. Orange and blue circles correspond to heated and control conditions, respectively. The gray area represents the marine heatwave period (d2-d10). Error bars indicate standard deviation of the mean.
(TIF)

## Acknowledgments

We would like to acknowledge the following individuals: Sébastien Mas, Rémi Valdès, Solenn Soriano, Kevin Mestre, Camille Suarez-Bazille, and David Parin of MEDIMEER, for their valuable help with the mesocosm setup, daily sampling, sensor installation, and dissolved nutrient analyses. We also wish to thank Emilie Eveque, Jean-François Thevenot, Luigia

Verde and the many other partners from the AQUACOSM Transnational Access program for their assistance with daily sampling and treatments.

## Author contributions

**Conceptualization:** Behzad Mostajir, Francesca Vidussi.

**Data curation:** Zoe Eglaine, Justine Courboulès, Francesco Cipolletta, Cécile Roques, Tanguy Soulié, Diana Sarno, Behzad Mostajir, Francesca Vidussi.

**Formal analysis:** Zoe Eglaine.

**Funding acquisition:** Behzad Mostajir, Francesca Vidussi.

**Investigation:** Zoe Eglaine, Justine Courboulès, Francesco Cipolletta, Cécile Roques, Tanguy Soulié, Diana Sarno, Behzad Mostajir, Francesca Vidussi.

**Project administration:** Behzad Mostajir, Francesca Vidussi.

**Resources:** Zoe Eglaine, Justine Courboulès, Francesco Cipolletta, Cécile Roques, Tanguy Soulié, Diana Sarno, Behzad Mostajir, Francesca Vidussi.

**Supervision:** Behzad Mostajir, Francesca Vidussi.

**Validation:** Zoe Eglaine, Behzad Mostajir, Francesca Vidussi.

**Visualization:** Zoe Eglaine.

**Writing – original draft:** Zoe Eglaine.

**Writing – review & editing:** Zoe Eglaine, Justine Courboulès, Francesco Cipolletta, Tanguy Soulié, Diana Sarno, Behzad Mostajir, Francesca Vidussi.

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
