## [Decision Letter · Decision Letter 0]

20 Aug 2025

Dear Dr. Eglaine,

Thank you for submitting your manuscript to PLOS ONE. After careful consideration, we feel that it has merit but does not fully meet PLOS ONE’s publication criteria as it currently stands. Therefore, we invite you to submit a revised version of the manuscript that addresses the points raised during the review process.

We look forward to receiving your revised manuscript.

Kind regards,

Rajdeep Roy

Academic Editor

PLOS ONE

Journal Requirements: 

Additional Editor Comments:

Dear Dr. Zoe

We have received two reviews and both have suggested minor revision. The authors are requested to go through the comments/ suggestion and submit the revised version. While revising please follow the journal guide lines.

Academic Editor

Reviewers' comments:

Reviewer's Responses to Questions

**Comments to the Author**

1. Is the manuscript technically sound, and do the data support the conclusions?

Reviewer #1: Yes

Reviewer #2: Yes

2. Has the statistical analysis been performed appropriately and rigorously?

Reviewer #1: Yes

Reviewer #2: Yes

3. Have the authors made all data underlying the findings in their manuscript fully available?

Reviewer #1: Yes

Reviewer #2: Yes

4. Is the manuscript presented in an intelligible fashion and written in standard English?

Reviewer #1: Yes

Reviewer #2: Yes

Reviewer #1: The main idea of the paper regarding to the understanding of each plankton web component responce to MHW was achieved. The idea is original and relevant to the field of studies. The methods were selected correctly and the results were provided in a sound maner. They were also well discussed. Regarding Conclusions, as this part is missing, the last abstract somehow summerise the said before. The References and Figures are appropriate.

I found some minor issues that may improve the manuscrip. Please consider them:

Line 143, you already mentioned shortening. Thus, you should not multiply it, as in the line 160 and others (l 262, table later…. Maybe better to live in a line 160 and delete an explanation of the shortening before.

Line 274, please correct ‘-‘sign for NO2 and NO3, it should be upper

Line 291, 294, 296, 297, 315, 316 etc for cells.ml, ‘.’ Should be in the middle of it to differentiate between dot and the multiply sign

Also, I noticed that sometimes you have Litters (L) and sometimes ml. It should be unified according to CI.

In line 247 you explained the shortening of PCA, please do it only once, there. It is not necessary to repeat as for example in line 415.

Line 614 species is repeated.

Please add to Line 586, what other groups you mean.

Reviewer #2: The present study "PONE-D-25-36140" by Eglaine at al. is a valuable contribution to understanding the effects of marine heatwaves on plankton communities in coastal ecosystems, specifically in the Thau Lagoon, In the Mediterranean Sea, Francia. Tha authors made a notable effort in preapring the mesocosmos experiments and, above all, in carryng out the sampling, as well as statistical analyses that support their finding. The research addresses a relevant topic in the context of climate change, using a solid detailed experimental approach. I believe the results are clear and consistent and support the proposed working hypotheses. Overall, the discussions are well-founded and reflect a careful analysis of the experimental data and their ecological context. Figures are of good quality. References are updated. Although I believe the current structure of the manuscript is adequate and has the potential to be published in PLOS-ONE, I think it could benefit that incluid from metazooplankton data: Although mentioned in the discussion, including more details on their dynamics and relationship with protozooplankton could enrich the analysis, as well as the impact on biogeochemical processes: Explore how changes in plankton communities affect processes such as the carbon cycle or oxygen production. Finally, conduct a thorough review to correct any grammatical or stylistic errors.

**Do you want your identity to be public for this peer review?** For information about this choice, including consent withdrawal, please see our Privacy Policy

Reviewer #1: No

Reviewer #2: No

---

## [Author Response · Author response to Decision Letter 1]

9 Sep 2025

Manuscript ID: PONE-D-25-36140

Title: Effects of a simulated marine heatwave on the structure and composition of Mediterranean plankton in a mesocosm study.

Dear Editor, Dear Reviewers,

We sincerely thank you for the time you have devoted to our manuscript and for your valuable comments. We have carefully considered all your remarks and revised the article accordingly. Below you will find a detailed response to each of the points raised.

You will find below a description of the layout changes made to highlight the changes made to the manuscript following comments.

- Changes and additions to the text are highlighted in yellow. Example: Adding text

- The parts or elements removed from the manuscript are crossed out and colored red. Example: text deletion

- Movement of sentences or paragraphs to another location in the revised version are indicated in green. Example: text movement

The numbers of lines mentioned below correspond to those of the “Revised Version Without Track Changes”.

Journal Requirements:

Requirement 1:

Response 1:

The manuscript was revised and supplementary files to comply with PLOS ONE’s style and file-naming requirements.

Requirement 2:

We note that the grant information you provided in the ‘Funding Information’ and ‘Financial Disclosure’ sections do not match.

Response 2:

The grant information was corrected so that it is consistent between the ‘Funding Information’ and ‘Financial Disclosure’ sections, and the correct grant numbers are now provided.

Requirement 3:

When completing the data availability statement of the submission form, you indicated that you will make your data available on acceptance. We strongly recommend all authors decide on a data sharing plan before acceptance, as the process can be lengthy and hold up publication timelines. Please note that, though access restrictions are acceptable now, your entire data will need to be made freely accessible if your manuscript is accepted for publication. This policy applies to all data except where public deposition would breach compliance with the protocol approved by your research ethics board. If you are unable to adhere to our open data policy, please kindly revise your statement to explain your reasoning and we will seek the editor's input on an exemption. Please be assured that, once you have provided your new statement, the assessment of your exemption will not hold up the peer review process.

Response 3:

The Data Availability Statement was updated and data underlying this study was already published in SEANOE (https://doi.org/10.17882/108308) and are now fully accessible, in accordance with PLOS ONE’s open data policy. DOI of published data was also mentioned in the Revised Version in Lines 215-216.

Requirement 4:

Please amend either the title on the online submission form (via Edit Submission) or the title in the manuscript so that they are identical.

Response 4:

The title was amended to ensure that it is identical in both the manuscript and the online submission form.

Requirement 5:

Response 5:

As there was no specific recommendation for additional citation was asked by the Reviewers, any change to the reference list was made.

Requirement 6:

Please review your reference list to ensure that it is complete and correct. If you have cited papers that have been retracted, please include the rationale for doing so in the manuscript text or remove these references and replace them with relevant current references. Any changes to the reference list should be mentioned in the rebuttal letter that accompanies your revised manuscript. If you need to cite a retracted article, indicate the article’s retracted status in the References list and also include a citation and full reference for the retraction notice.

Response 6:

We have carefully checked and revised the reference list to ensure completeness and accuracy. No retracted articles remain cited.

Reviewers’ comments:

Reviewer #1 – comment 1:

“Line 143, you already mentioned shortening. Thus, you should not multiply it, as in the line 160 and others (l 262, table later…. Maybe better to live in a line 160 and delete an explanation of the shortening before.”

Reviewer #1 – response to comment 1:

Thank you for this suggestion. We agree with this comment and remove the explanation of the abbreviation at lines 143, 262, and 281, (lines 143, 265, and 284 in revised version) keeping it only at line 160 as recommended. This modification avoids redundancy and improves the clarity of the manuscript.

Reviewer #1 – comment 2:

“Line 274, please correct ‘-‘ sign for NO2 and NO3, it should be upper.”

Reviewer #1 – response to comment 2:

Thanks for this comment. The notations of NO2 and NO3 were corrected by placing the minus sign (“-”) at the upper position (NO2- and NO3- ) in line 277.

Reviewer #1 – comment 3:

“Line 291, 294, 296, 297, 315, 316 etc for cells.ml, ‘.’ Should be in the middle of it to differentiate between dot and the multiply sign.

Also, I noticed that sometimes you have Litters (L) and sometimes ml. It should be unified according to CI.”

Reviewer #1 – response to comment 3:

Thank you for this remark. As suggested, we have replaced the dot (“.”) with the middle dot (“·”) in all relevant expressions, so that “cells.ml-1” now appears as “cells·mL-1” (lines 294, 297-301, 314, 318, 319, 321, 325-327, 356, 360, 376, 377, and 380).

Following the SI unit’s brochure, we have throughout the manuscript also

• Standardized volume units to “mL” (instead of “L” or “ml”),

• Harmonized the notation “cells·mL-1”, and

• Added appropriate spacing between numbers and units/symbols (e.g., °C, %, <, >, and =).

Reviewer #1 – comment 4:

“In line 247 you explained the shortening of PCA, please do it only once, there. It is not necessary to repeat as for example in line 415.”

Reviewer #1 – response to comment 4:

As suggested, the repeated explanation was removed at line 418 and retained the definition only at line 250 to avoid redundancy.

Reviewer #1 – comment 5:

“Line 614 species is repeated.”

Reviewer #1 – response to comment 5:

The redundancy was corrected by removing the repeated word species at line 634.

Reviewer #1 – comment 6:

“Please add to Line 586, what other groups you mean.”

Reviewer #1 – response to comment 6:

Thanks for this suggestion. To clarify this, the groups referred to “other groups” were specified in lines 586. The revised sentence now reads:

Line 605. “Cyanobacteria, picophytoplankton, diatom, autotrophic, and mixotrophic flagellates showed the highest positive resilience slopes compared to the other groups (viruses, bacteria, nanophyto, HNF, all trophic modes dinoflagellates, and ciliates) that exhibited negative or near-zero slopes.”

Reviewer #2 – comment 1:

“I think it could benefit that incluid from metazooplankton data: Although mentioned in the discussion, including more details on their dynamics and relationship with protozooplankton could enrich the analysis, as well as the impact on biogeochemical processes: Explore how changes in plankton communities affect processes such as the carbon cycle or oxygen production.”

Reviewer #2 – response to comment 1:

We fully agree with this point. Accordingly, we expanded the discussion by including more information on mesozooplankton dynamic, their interactions within food web, and their influence on biogeochemical cycles. Two new paragraphs therefore were added to the Revised Version:

• Lines 504 – 511:

“Additionally, Zervouldaki et al. (2024) reported that the copepods, polychaetes, larvae of bivalve, and gastropods, were the dominant mesozooplankton groups present in the mesocosms during the same experiment. The MHW condition had a significant positive effect on copepods, mainly driven by an increase in harpacticoid copepods, while a positive trend was also observed for polychaete larvae [31]. These results are consistent with previous studies demonstrating that warming can stimulate mesozooplankton development at the same location [37] and enhance zooplankton grazing in the Baltic sea [66]. Collectively, these findings highlight that the decrease in the abundances of protozooplankton and phytoplankton communities may be mediated by increased predation pressure exerted by mesozooplankton. “

• Lines 525 – 532:

“In addition, Soulié et al. (2022) reported that under MHW, the system tended to shift toward heterotrophy due to increased respiration and phytoplankton mortality [30], further supporting the idea that mesozooplankton exerted a strong top-down control on diatoms and protozooplankton under MHW condition. These outcomes could also impact biogeochemical cycles, particularly the oxygen cycle, where oxygen consumption may exceed production under MHW, with cascading consequences for the carbon cycle through an intensified release of carbon into the water. Consequently, under MHW, coastal Mediterranean systems may act as a carbon source to the atmosphere rather than as a sink of carbon.”

These additions strengthen the discussion by clarifying how food web changes can affect biogeochemical process and overall ecosystem functioning.

Reviewer #2 – comment 3:

“Finally, conduct a thorough review to correct any grammatical or stylistic errors.”

Reviewer #2 – response to comment 3:

The entire manuscript was carefully revised to correct grammatical and stylistic errors. Additionally, we would like to point out that the entire manuscript has been reviewed by Editage, a recognized brand in the English edition (https://www.editage.com/) to ensure linguistic and grammatical accuracy prior to submission (an available certificate was issued by Editage).

We would like to thank the editor and reviewers once again for carefully evaluating our work. We believe that the revisions address the concerns raised and strengthen the manuscript. We look forward to your further consideration.

Sincerely,

Zoé Eglaine

---

## [Decision Letter · Decision Letter 1]

22 Oct 2025

Dear Dr. Eglaine,

Thank you for submitting your manuscript to PLOS ONE. After careful consideration, we feel that it has merit but does not fully meet PLOS ONE’s publication criteria as it currently stands. Therefore, we invite you to submit a revised version of the manuscript that addresses the points raised during the review process.

We look forward to receiving your revised manuscript.

Kind regards,

Rajdeep Roy

Academic Editor

PLOS ONE

Journal Requirements:

Reviewers' comments:

Reviewer's Responses to Questions

**Comments to the Author**

Reviewer #1: All comments have been addressed

Reviewer #2: All comments have been addressed

2. Is the manuscript technically sound, and do the data support the conclusions?

Reviewer #1: Yes

Reviewer #2: Yes

3. Has the statistical analysis been performed appropriately and rigorously?

Reviewer #1: Yes

Reviewer #2: Yes

4. Have the authors made all data underlying the findings in their manuscript fully available?

Reviewer #1: Yes

Reviewer #2: Yes

5. Is the manuscript presented in an intelligible fashion and written in standard English?

Reviewer #1: Yes

Reviewer #2: Yes

Reviewer #1: Dear authors,

Thank you for taking on board all my comments.

One additional point was noticed: line 219. Please rewrite it so that it does not imply that picoplankton cannot include cyanobacteria, for example. It usually consists of lots of cyanobacteria as well as other groups, but the way it is written implies that it is something completely different.

My suggestion would be:

Resistance, resilience, recovery, and temporal stability were calculated according to the methods described by Hillebrand et al. [24] for viruses, bacteria, cyanobacteria, diatoms, dinoflagellates, flagellates for auto-, hetero- and mixotrophic, undetermined groups, HNF, and ciliates. Additionally, a distinct group of very small algal cells, referred to as picophytoplankton and nanophytoplankton were separated.

Also, on line 168, you wrote 'prokaryote' and 'eukaryote'. Please change these to 'prokaryotic' and 'eukaryotic'.

After it manuscript can be published.

Reviewer #2: The authors responded well to the reviewers' questions and comments. They addressed each point and made the changes needed to meet the editor's and reviewers' suggestions.

Among the actions taken, the following stand out:

Correction of redundancies and improvement of clarity:

The authors removed repeated explanations and adjusted the presentation of abbreviations and notations to avoid redundancies and improve the overall clarity of the text.

Correction of technical errors:

They corrected the placement of symbols in chemical formulas and standardized measurement units according to the reviewers’ recommendations.

Expansion of the discussion:

The authors added more information about mesozooplankton dynamics and how these affect biogeochemical processes like the carbon and oxygen cycles. This made the discussion and analysis more complete.

Linguistic revision:

The authors carefully reviewed the language to fix grammar and style issues, which improved the manuscript's quality and readability.

Compliance with editorial requirements:

They also updated the title, fixed the funding information, and revised the data availability statement to follow the journal's editorial policies.

Overall, the authors worked hard to improve the manuscript and responded well to the reviewers' feedback. These changes have made the work stronger and helped it move toward publication.

In summary, the manuscript presents a solid experimental design, detailed analyses, and consistent results that support the conclusions. Moreover, the ecological implications discussed are relevant and well-grounded in data and scientific literature.

Conclusion: Based on the revisions made and the manuscript's overall quality, I consider it suitable for publication.

**Do you want your identity to be public for this peer review?** For information about this choice, including consent withdrawal, please see our Privacy Policy

Reviewer #1: No

Reviewer #2: No

---

## [Author Response · Author response to Decision Letter 2]

23 Oct 2025

Manuscript ID: PONE-D-25-36140R1

Title: Effects of a simulated marine heatwave on the structure and composition of Mediterranean plankton in a mesocosm study.

Dear Editor, Dear Reviewers,

We sincerely thank you again for the time and efforts that you dedicated to reviewing our manuscript and for your valuable and constructive comments. We have carefully considered all your remarks and revised the article accordingly. Below, we provide a detailed, point by point response to each of the reviewers’ comments.

We also describe the layout modifications made to clearly highlight the changes introduced in the second revised version of the manuscript in yellow (example: Adding text), while parts or elements removed from the manuscript are crossed out and colored red (example: text deletion).

The numbers of lines mentioned below correspond to those of the “Second Revised Version Without Track Changes”.

Journal Requirements:

Requirement 1:

Response 1:

As no specific recommendation for additional citations was made by the Reviewers, no changes were made to the reference list.

Requirement 2:

Please review your reference list to ensure that it is complete and correct. If you have cited papers that have been retracted, please include the rationale for doing so in the manuscript text or remove these references and replace them with relevant current references. Any changes to the reference list should be mentioned in the rebuttal letter that accompanies your revised manuscript. If you need to cite a retracted article, indicate the article’s retracted status in the References list and also include a citation and full reference for the retraction notice.

Response 2:

We have carefully reviewed and updated the reference list to ensure its completeness and accuracy. All references were verified, and no retracted articles remain cited.

Reviewers’ comments:

Reviewer #1 – comment 1:

One additional point was noticed: line 219. Please rewrite it so that it does not imply that picoplankton cannot include cyanobacteria, for example. It usually consists of lots of cyanobacteria as well as other groups, but the way it is written implies that it is something completely different.

My suggestion would be: Resistance, resilience, recovery, and temporal stability were calculated according to the methods described by Hillebrand et al. [24] for viruses, bacteria, cyanobacteria, diatoms, dinoflagellates, flagellates for auto-, hetero- and mixotrophic, undetermined groups, HNF, and ciliates. Additionally, a distinct group of very small algal cells, referred to as picophytoplankton and nanophytoplankton were separated.

Reviewer #1 – response to comment 1:

We fully agree with this point and rewrite the sentence as suggested by Reviewer 1.

Lines 218 – 221:

“Resistance, resilience, recovery, and temporal stability were calculated according to the methods described by Hillebrand et al. [24] for viruses, bacteria, cyanobacteria, diatoms, dinoflagellates, flagellates for auto-, hetero- and mixotrophic, undetermined groups, HNF, and ciliates. Additionally, a distinct group of very small algal cells, referred to as picophytoplankton and nanophytoplankton were separated”

Reviewer #1 – comment 2:

On line 168, you wrote 'prokaryote' and 'eukaryote'. Please change these to 'prokaryotic' and 'eukaryotic'.

Reviewer #1 – response to comment 2:

We have replaced ‘prokaryote’ and ‘eukaryote’ by ‘prokaryotic’ and ‘eukaryotic’ in line 168.

Sincerely,

Zoé Eglaine and co-authors

---

## [Decision Letter · Decision Letter 2]

5 Nov 2025

Effects of a simulated marine heatwave on the structure and composition of Mediterranean plankton in a mesocosm study

PONE-D-25-36140R2

Dear Dr. Eglaine

We’re pleased to inform you that your manuscript has been judged scientifically suitable for publication and will be formally accepted for publication once it meets all outstanding technical requirements.

Kind regards,

Rajdeep Roy

Academic Editor

PLOS ONE

Reviewers' comments:

Reviewer's Responses to Questions

**Comments to the Author**

Reviewer #1: All comments have been addressed

2. Is the manuscript technically sound, and do the data support the conclusions?

Reviewer #1: Yes

3. Has the statistical analysis been performed appropriately and rigorously?

Reviewer #1: Yes

4. Have the authors made all data underlying the findings in their manuscript fully available?

Reviewer #1: Yes

5. Is the manuscript presented in an intelligible fashion and written in standard English?

Reviewer #1: Yes

Reviewer #1: The manuscript is accepted from my side, but a few remarks left.

Please consider typo on a page 14 (line 352), there C. is correct but C. closterium. Species name should start from lower case.

**Do you want your identity to be public for this peer review?** For information about this choice, including consent withdrawal, please see our Privacy Policy

Reviewer #1: No

---

## [Editor Report · Acceptance letter]

PONE-D-25-36140R2

PLOS ONE

Dear Dr. Eglaine,

I'm pleased to inform you that your manuscript has been deemed suitable for publication in PLOS ONE. Congratulations! Your manuscript is now being handed over to our production team.

Kind regards,

on behalf of

Dr. Rajdeep Roy

Academic Editor

PLOS ONE